# Maternal and Fetal Outcomes after Prior Mid-Trimester Uterine Rupture: A Systematic Review with Our Experience

**DOI:** 10.3390/medicina57121294

**Published:** 2021-11-24

**Authors:** Shinya Matsuzaki, Tsuyoshi Takiuchi, Takeshi Kanagawa, Satoko Matsuzaki, Misooja Lee, Michihide Maeda, Masayuki Endo, Tadashi Kimura

**Affiliations:** 1Department of Gynecology, Osaka International Cancer Institute, Osaka 541-8567, Japan; shinya.matsuzaki@oici.jp (S.M.); maeda.rf@gmail.com (M.M.); 2Department of Obstetrics and Gynecology, Graduate School of Medicine, Osaka University, Osaka 565-0871, Japan; tadashi@gyne.med.osaka-u.ac.jp; 3Department of Clinical Genomics, Graduate School of Medicine, Osaka University, Osaka 565-0871, Japan; 4Department of Maternal Fetal Medicine, Osaka Medical Center and Research Institute for Maternal and Child Health, Izumi 594-1101, Japan; takekana@wch.opho.jp; 5Osaka General Medical Center, Department of Obstetrics and Gynecology, Osaka 558-8558, Japan; satoko_tsuru@yahoo.co.jp; 6Department of Obstetrics and Gynecology, Shibata Pediatrics Clinic, San Francisco, CA 94118, USA; misooja810@gmail.com; 7Department of Health Science, Osaka University Graduate School of Medicine, Osaka 565-0871, Japan; mendoh@sahs.med.osaka-u.ac.jp

**Keywords:** mid-trimester, second trimester, placenta accreta spectrum, uterine rupture

## Abstract

*Background and Objectives*: Since spontaneous uterine rupture in the mid-trimester is rare, maternal and fetal outcomes in subsequent pregnancies remain unclear. Therefore, this study aimed to examine the maternal and fetal outcomes of subsequent pregnancies after prior mid-trimester uterine rupture. *Materials and Methods*: A systematic review using PubMed, the Cochrane Central Register of Controlled Trials, and Scopus until 30 September 2021, was conducted in compliance with the Preferred Reporting Items for Systematic Reviews and Meta-Analyses guidelines. The studies that clarified the maternal and fetal outcomes after prior mid-trimester uterine rupture and our case (*n* = 1) were included in the analysis. *Results*: Among the eligible cases, there were five women with eight subsequent pregnancies after prior mid-trimester uterine rupture. The timing of prior mid-trimester uterine rupture ranged from 15 to 26 weeks of gestation. The gestational age at delivery in subsequent pregnancies was 23–38 gestational weeks. Among the included cases (*n* = 8), those involving prior mid-trimester uterine rupture appeared to be associated with an increased prevalence of placenta accreta spectrum (PAS) (*n* = 3, 37.5%) compared with those involving term uterine rupture published in the literature; moreover, one case exhibited recurrent uterine rupture at 23 weeks of gestation (12.5%). No maternal deaths have been reported in subsequent pregnancies following prior mid-trimester uterine rupture. Fetal outcomes were feasible, except for one pregnancy with recurrent mid-trimester uterine rupture at 23 weeks of gestation, whose fetus was alive complicated by cerebral palsy. *Conclusions*: Our findings suggest that clinicians should be aware of the possibility of PAS and possible uterine rupture in pregnancies after prior mid-trimester uterine rupture. Further case studies are warranted to assess maternal and fetal outcomes in pregnancies following prior mid-trimester prior uterine rupture.

## 1. Introduction

Uterine rupture is potentially catastrophic for the mother and fetus [1,2], with an incidence of 5.9/10,000, and predominantly occurs during the intrapartum period and rarely before labor [1,3]. In developed countries, the primary risk factor for uterine rupture is prior cesarean delivery, and the estimated incidence of rupture is higher in women who undergo a trial of labor after cesarean delivery (TOLAC) than in those who undergo planned repeat cesarean delivery (PRCD) (0.78% with TOLAC versus 0.02% with PRCD) [4]. Another major risk factor for uterine rupture is previous uterine surgery, such as laparoscopic or abdominal myomectomy [5].

Notably, prior uterine rupture may be a significant risk factor for uterine rupture in subsequent pregnancies [6,7]. The recurrence-risk range is 0–33% in previous reports, most of which are case reports and reviews [7,8,9]. A study published in 2020 (*n* = 37) demonstrated no recurrent uterine rupture in subsequent pregnancies after prior uterine rupture in cases where cesarean delivery was performed between 36 and 37 weeks of gestation [6]. However, as the incidence of spontaneous uterine rupture in the mid-trimester is rare, the rate of recurrent uterine rupture in subsequent pregnancies remains unclear.

A previous study reported a woman with placenta accreta spectrum (PAS) in subsequent pregnancies after mid-trimester uterine rupture [10]. In women with PAS, myometrial invasion interferes with placental detachment, leading to significant maternal morbidity [11,12,13,14,15]. Postpartum hemorrhage is a major complication of PAS, and intraoperative blood loss potentially exceeds several liters, possibly leading to shock and coagulopathy [16,17,18,19,20,21,22]. There is a risk of injury to surrounding organs with PAS, particularly to the bladder and ureter. Therefore, cesarean hysterectomy without an attempt to remove the placenta is often required to avoid these complications. Ultimately, there is an increased risk of maternal mortality with PAS [17,23,24,25,26].

We conducted a systematic literature review to determine the maternal and fetal outcomes of subsequent pregnancies after mid-trimester uterine rupture. We also report a case of a woman with PAS following spontaneous uterine rupture at 15 weeks’ gestation, and this case was included in the analysis.

## 2. Materials and Methods

### 2.1. Systematic Literature Review Approach

A systematic review was performed to analyze maternal and fetal outcomes after prior mid-trimester uterine rupture, as previously described [27,28,29]. The outcomes of interest were as follows: the rate of uterine rupture, PAS, and maternal outcomes (hysterectomy, and maternal death). A systematic literature search was conducted using PubMed, Scopus, and the Cochrane Central Register of Controlled Trials (CENTRAL) from inception to 30 September 2021, according to the Preferred Reporting Items for Systematic Reviews and Meta-Analyses (PRISMA) guidelines, 2020 edition [30]. The protocol of systematic review has not been registered and we did not use any online tool in this process.

### 2.2. Eligibility Criteria, Information Sources, and Search Strategy

The terms “prior uterine rupture at mid-trimester” and “subsequent pregnancy” were searched using the text words listed in Appendix A. In this study, “mid-trimester” was defined as the period spanning the 15th through the 28th completed week (99 to 196 days) of gestation.

### 2.3. Study Selection

The inclusion criteria, which were based on the Patient/Population, Intervention, Comparison, Outcome, and Study (PICOS) process, are shown in Appendix A. Studies were selected according to the following inclusion criteria: (1) occurrence of prior uterine rupture in the mid-trimester, (2) occurrence of complete uterine rupture, and (3) clarification of maternal and fetal outcomes in subsequent pregnancies.

The exclusion criteria were as follows: (1) insufficient information regarding the outcomes of interest, (2) ectopic pregnancy, (3) uterine rupture due to uterine anomaly, (4) congenital disorders, such as Ehlers–Danlos type IV (5) articles not published in English, and (6) conference abstracts.

Studies were identified by screening the titles, abstracts, and full texts of the relevant articles. Titles and abstracts were independently screened by two authors (S.M. (Shinya Matsuzaki) and S.M. (Satoko Matsuzaki)).

### 2.4. Data Extraction

One of the authors (S.M. (Shinya Matsuzaki)) extracted the data and recorded the following variables: year of study, first author’s name, study location, number of included cases, PAS and uterine rupture definitions, subsequent obstetric outcomes (rates of uterine rupture and PAS), and maternal outcomes (hysterectomy, and maternal death). The review author (S.M. (Satoko Matsuzaki)) double-checked the data.

## 3. Results

### 3.1. Case Presentation

A 34-year-old, gravida 5, para 2, woman was referred to our hospital at 13 weeks’ gestation for high-risk pregnancy management. The patient had a history of appendectomy, right ovarian cystectomy, and cervical conization. The patient’s obstetric history was as follows: first pregnancy, term vaginal delivery; second pregnancy, lower transverse cesarean section at 36 weeks’ gestation; third pregnancy, spontaneous uterine rupture of a previous transverse scar at 15 weeks’ gestation (detailed information has been previously reported) [31]. Two-and-a-half years after the third pregnancy, the patient visited the community hospital with regard to a fourth pregnancy. Transvaginal ultrasonography at 8 weeks’ gestation revealed a gestational sac in the lower uterine segment (Figure 1a). Therefore, the patient was referred to our hospital due to the risk of uterine rupture.

At the patient’s first visit in gestational week 13, transvaginal ultrasonography revealed a single viable intrauterine fetus with appropriate growth and restricted space between the placenta and bladder (Figure 1b). To evaluate the cause of prior uterine rupture in the mid-trimester, trauma or congenital/acquired weakness of the myometrium was evaluated as follows: (i) ruling out uterine anomaly; (ii) exclusion of congenital disorders, such as Ehlers–Danlos type IV; (iii) confirmation that the previous uterine scar was a lower uterine incision for cesarean delivery; and (iv) confirmation of no trauma before uterine rupture. While placenta previa was excluded by transvaginal ultrasonography, magnetic resonance imaging and transabdominal ultrasonography indicated placenta increta without uterine dehiscence (Figure 1c,d). The patient was informed of the risk of maternal morbidity, specifically PAS, and chose to undergo cesarean hysterectomy without an attempt to remove the placenta at 34 weeks of gestation. The patient was well-educated regarding the symptoms that provide an early indication of uterine rupture.

The patient was admitted to our hospital for suspected placenta increta at a gestational age of 33 weeks 6 days and underwent an elective cesarean delivery with hysterectomy on the following day. Since the patient declined fertility preservation, we performed a transverse uterine fundal incision, as previously reported [32,33], and a healthy female infant with a birth weight of 1820 g was successfully delivered (Apgar scores of 8 and 9 at 1 and 5 min, respectively). The placenta and enlarged vessels could be identified through the thin uterine wall (Figure 2a,b). Total intraoperative blood loss was 1430 mL; hence, the patient received four units of packed red blood cells. Histological examination of the surgical specimen revealed placenta increta (Figure 2c,d). The infant experienced transient tachypnea of the newborn but recovered well, and both mother and baby were eventually discharged.

### 3.2. Study Selection for Systematic Review

Figure 3 illustrates the study selection scheme. Overall, 96 studies were examined; among these, four studies involving eight deliveries with prior mid-trimester uterine rupture fulfilled the inclusion criteria for analysis [8,10,27,28].

### 3.3. Study Characteristics

Table 1 shows a summary of evaluated studies (*n* = 4), including our case [8,10,34,35]. Among these studies, three were case reports [10,34,35], and one was an original article that examined the effects of 37 prior uterine ruptures on maternal outcomes in subsequent pregnancies. In the original article, there was one case of prior uterine rupture in gestational week 17 [8].

### 3.4. Maternal and Fetal Outcomes

In the included studies, there were five women with eight pregnancies after prior mid-trimester uterine rupture. Of these women, the timing of uterine rupture was between 15 and 26 weeks of gestation, and the causes of uterine rupture were as follows: unidentified in two cases, prior cesarean delivery in two, prior myomectomy in one, and not applicable in one. The uterine rupture sites were the fundus in three women and prior uterine cesarean scar in two. The type of repair performed for uterine rupture was clarified in three of five cases (two three-layer and one double-layer repair). None of the studies mentioned whether interrupted or continuous sutures were used.

Among the eight pregnancies, three were complicated by PAS and one by recurrent mid-trimester uterine rupture. Among the women with PAS, two developed placenta percreta and the other placenta increta. The site of PAS was similar to that of the prior uterine rupture site in all instances, and all women underwent cesarean hysterectomy. No maternal deaths were reported in subsequent pregnancies following prior mid-trimester uterine rupture. Fetal outcomes were feasible, except for one pregnancy with recurrent mid-trimester uterine rupture at 23 weeks of gestation, in which the fetus was alive, though complicated by cerebral palsy.

## 4. Discussion

### 4.1. Key Findings

Although the number of included cases is limited, this study yielded the following three key findings: (i) the rate of PAS was high in subsequent pregnancies after prior mid-trimester uterine rupture, (ii) all women with PAS after mid-trimester uterine rupture underwent cesarean hysterectomy due to severe PAS, (iii) recurrent uterine rupture was observed in one of eight pregnancies, and (iv) maternal and fetal outcomes after prior mid-trimester uterine rupture have been understudied due to its rarity.

### 4.2. Strengths and Limitations

This is the first systematic review to examine the maternal outcomes of subsequent pregnancies after mid-trimester uterine rupture. Our findings suggest that clinicians should focus on the high rate of PAS and uterine rupture in pregnancies after prior mid-trimester uterine rupture. However, this study had notable limitations. First, bias was not measured because all the included studies involved a single case. Second, we only identified five women and eight subsequent pregnancies after mid-trimester uterine rupture. Therefore, our study does not possess sufficient power to draw a solid conclusion regarding the maternal outcomes of subsequent pregnancies after prior mid-trimester uterine rupture. Nevertheless, as no systematic review on the topic at hand has yet been conducted, we believe that our study is beneficial.

Third, a previous systematic review found that pre-labor uterine rupture was associated with PAS in approximately half of subsequent pregnancies. Since the risk of recurrent PAS is approximately 20% [36,37], the possible effect of recurrent PAS after prior mid-trimester uterine rupture should be acknowledged. Fourth, publication bias is a matter of concern because cases of favorable outcomes in subsequent pregnancies after prior mid-trimester uterine rupture might not have been reported.

### 4.3. Comparison with Existing Literature

#### 4.3.1. Maternal and Fetal Outcomes after Uterine Rupture

Although uterine rupture is potentially catastrophic for the mother and fetus, recent reports suggest that subsequent pregnancy outcomes in patients with prior uterine rupture are apparently favorable if managed in a standard manner [6,7,9]. In a previous retrospective study, the maternal outcomes in 37 women with 59 pregnancies and prior uterine rupture were examined. In the foregoing study, cesarean delivery at approximately 36–37 weeks of gestation was performed [6]. Of the included cases (*n* = 59), no uterine rupture was observed, and one case of PAS was observed. This study concluded that the maternal outcomes of women with prior uterine rupture was favorable, provided they underwent cesarean delivery at 36–37 weeks of gestation.

A recent systematic review examined the maternal outcomes of pre-labor uterine rupture between 14 and 34 weeks of gestation using the PubMed, Google Scholar, and Embase databases from 1988 to 2020 [38]. The systematic review included 80 singleton intrauterine pregnancies, and the mean gestational week of uterine rupture was approximately 22 weeks. Among these cases (*n* = 80), the causes of uterine rupture were as follows: previous cesarean deliveries (*n* = 36; 45%), myomectomy (*n* = 20; 25%), uterine malformations (*n* = 13; 16.3%), previous uterine rupture (*n* = 8; 10%), and previous classical uterine incision (*n* = 6; 7.5%). This systematic review revealed that 35 cases (43.8%) were associated with PAS. This study did not examine the outcomes of subsequent pregnancy.

Although possible publication bias should be acknowledged, this study yielded the following two important findings: (i) early uterine rupture was associated with a scarred uterus (*n* = 62; 77.5%) and (ii) PAS was correlated with early pre-labor uterine rupture (*n* = 35; 43.8%). These findings suggest that the etiology of mid-trimester uterine rupture is different from that of term uterine rupture, as most term uterine ruptures in resource-rich countries are associated with a TOLAC delivery [39].

#### 4.3.2. Maternal and Fetal Outcomes after Mid-Trimester Uterine Rupture

The maternal outcomes of subsequent pregnancies after prior uterine rupture have been understudied, and all studies included at least one case of mid-trimester uterine rupture. In the current study, the rates of PAS and recurrent uterine rupture were 37.5% and 12.5%, respectively. Although the data were limited, and the possibility of severe publication bias should be acknowledged, the rates of PAS and uterine rupture after prior mid-trimester uterine rupture were possibly higher than those after prior term uterine rupture. Further studies are warranted to reveal the maternal outcomes of subsequent pregnancies after prior mid-trimester uterine rupture.

The high rate of PAS after mid-trimester uterine rupture has not been discussed. In a previous retrospective study that examined maternal outcomes after term uterine rupture, one of 37 women was complicated by PAS. However, the current study found three of eight pregnancies to be complicated by PAS after prior mid-trimester uterine rupture. This inconsistency may be explained by the difference in uterine rupture site. In general, the site of uterine rupture at term is determined by the previous transverse uterine scar, and over half of mid-trimester uterine ruptures occur at the uterine fundus (Table 1). We hypothesized that uterine rupture at the fundus possibly leads to a high rate of PAS.

#### 4.3.3. Timing of Delivery after Mid-Trimester Uterine Rupture

Optimal timing of delivery after mid-trimester uterine rupture is an essential discussion topic; however, we could not discuss this matter due to limited data. While this is merely our opinion, based on the recommended timing of delivery in women with PAS of 34–35 gestational weeks [17], the timing of delivery in women with PAS after mid-trimester uterine rupture may be recommended at 34–35 weeks of gestation. However, the timing of delivery in women without PAS after mid-trimester uterine rupture is unknown.

A previous study suggested that elective cesarean delivery at 36–37 weeks of gestation potentially leads to favorable maternal and fetal outcomes after uterine rupture [6]. Further studies that examine the timing of delivery after prior mid-trimester uterine rupture are warranted.

#### 4.3.4. Repair of Uterine Rupture

Three of the five included studies mentioned how repair of uterine rupture was performed, with all cases undergoing multiple-layer suturing. This systematic review found that no studies have reported the rates of uterine rupture in subsequent pregnancy at which different approaches to repair of uterine repair are adopted. With regard to the uterine closure following cesarean delivery, a recent systematic review that included nine randomized controlled trials with 3969 women compared the myometrial thickness between single-layer and double-layer sutures [40]. In this study, women who received single-layer closure were more likely to have a thinner residual myometrium on ultrasound compared to those who received double-layer closure [40]. This suggests that multiple-layer sutures may be preferred for the repair of mid-trimester uterine rupture.

Successful repair of mid-trimester uterine rupture with continuation of the pregnancy until gestational week 32 has been reported [41]. In this case, massive hemoperitoneum following uterine rupture at gestational week 20 was observed. To repair the uterine rupture, the authors inserted three pulley stitches from the pericervical muscular tissue to the myometrium above the ruptured area and placed a semisynthetic mesh at the rupture site. This allowed the patient to reach gestational week 32 and successfully deliver a live newborn through the cesarean route. Future studies to determine the optimal surgical technique for prevention of recurrence after mid-trimester uterine rupture are warranted.

### 4.4. Discussion of Our Case

In our case, the patient had a history of cesarean delivery, and the cause of mid-trimester uterine rupture appeared to be previous uterine cesarean scar. Since mid-trimester uterine rupture due to a previous cesarean scar is rare, we ruled out the risk of uterine rupture due to an unscarred uterus. Rupture of an unscarred pregnant uterus is a rare event, and its estimated occurrence rate is between 0.005% and 0.02% [42,43].

To assess the risk of uterine rupture in an unscarred uterus, it is recommended that the following conditions are verified [16,42,43,44]: (i) uterine malformation; (ii) a naturally weak myometrium due to a congenital disorder, such as Ehlers–Danlos type IV; (iii) trauma, such as vehicle crashes and obstetric maneuvers; and (iv) other factors (exposure to uterotonic drugs, high parity, advanced maternal age, prolonged labor, multiple gestations, etc.). Since our patient did not meet any of the criteria for uterine rupture in an unscarred uterus, we considered the prior cesarean delivery to be the cause of uterine rupture in the mid-trimester.

Prior mid-trimester uterine rupture and PAS are risk factors for uterine rupture, and our patient had a high risk of uterine rupture. The optimal management of such a patient during pregnancy remains unknown; thus, clinicians and patients should be aware of the following symptoms of uterine rupture: abnormal fetal heart tracing [45,46], abdominal pain with or without hemodynamic changes [47], cessation of uterine contractions, and antepartum vaginal bleeding.

### 4.5. Conclusions and Implications

#### 4.5.1. Implications for Practice

Our study demonstrates that the rates of PAS and uterine rupture are potentially high in pregnancies after prior mid-trimester uterine rupture. Therefore, these findings will help clinicians to be more attentive to PAS and uterine rupture during subsequent pregnancies after mid-trimester uterine rupture. Nevertheless, future studies that include a large number of women with prior mid-trimester uterine rupture are necessary to confirm our findings.

#### 4.5.2. Implications for Research

If the rate of recurrent uterine rupture is high after a prior mid-trimester uterine rupture, an examination of the optimal timing of delivery is imperative. As randomized controlled trials or prospective studies are difficult to conduct due to the rarity of mid-trimester uterine rupture, large-scale retrospective studies are warranted to determine the rate of uterine rupture and optimal timing of delivery after prior mid-trimester uterine rupture.

## Figures and Tables

**Figure 1 medicina-57-01294-f001:**
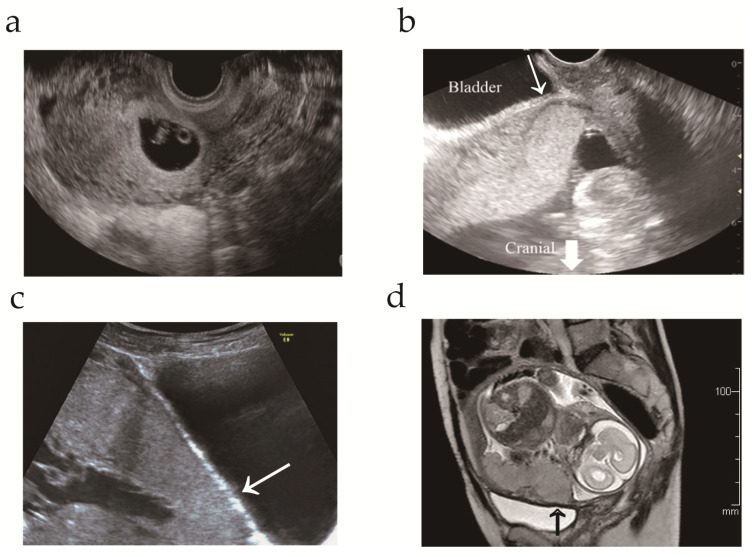
Transvaginal and magnetic resonance imaging during pregnancy. (**a**) Transvaginal ultrasonogram at 8 weeks’ gestation revealing a gestational sac located in the lower uterine segment; (**b**) Transvaginal ultrasonogram at 13 weeks’ gestation revealing decreased space (arrow) between the placenta and bladder; (**c**) Transabdominal ultrasonogram at 33 weeks’ gestation. The arrow indicates the loss of the normal hypoechoic retroplacental zone; (**d**) T2 magnetic resonance image at 25 weeks’ gestation. The arrow indicates the loss of the thin, dark subplacental myometrium zone.

**Figure 2 medicina-57-01294-f002:**
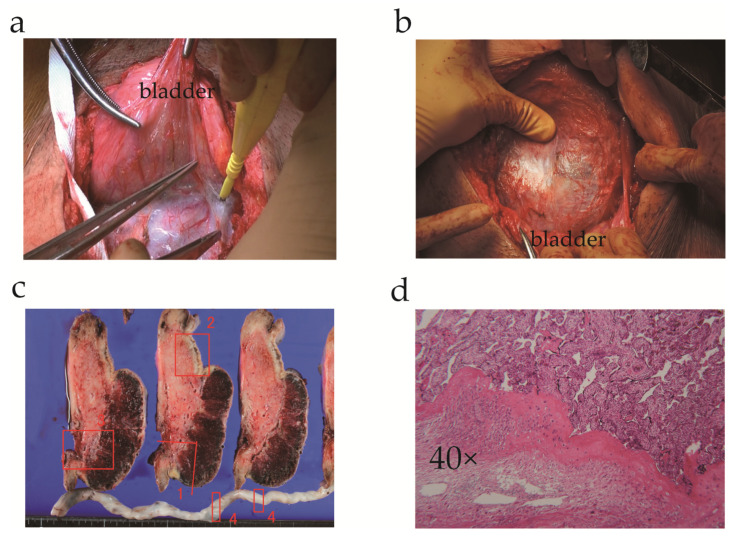
Intraoperative images and histopathological examination (**a**,**b**) Intraoperative picture revealing enlarged vessels and placenta through the thin anterior uterine wall after separating the bladder from the uterus; (**c**) The cut section of the hysterectomy specimen, showing placenta increta in section numbers 1 and 3 (number 2: uterine myometrium without placenta; number 4: cord vessels); (**d**) Histopathological image of uterine tissue revealing chorionic villi invading the myometrium.

**Figure 3 medicina-57-01294-f003:**
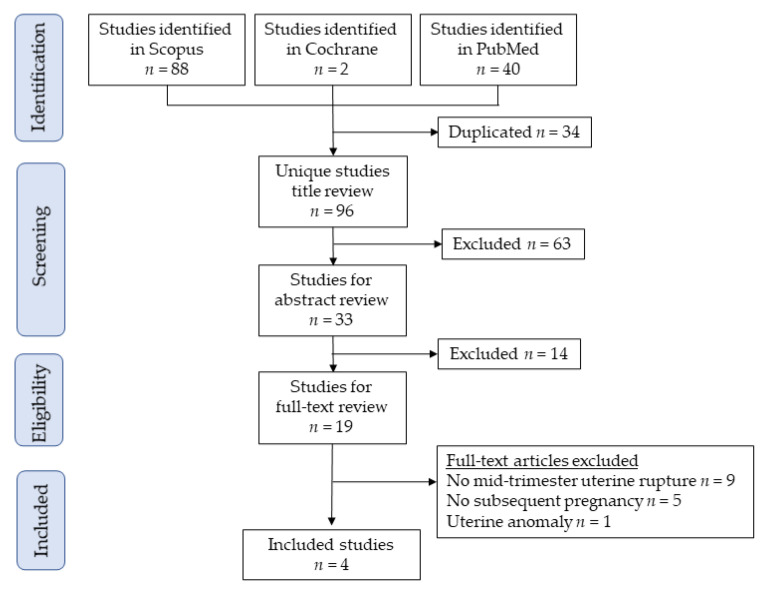
Study selection scheme used for the systematic review of the literature.

**Table 1 medicina-57-01294-t001:** Summary of the systematic review of literature regarding subsequent pregnancies after prior mid-trimester uterine rupture.

Author	Year	Timing	Site	No.	Cause	GA	Maternal Outcome	Site of PAS	Fetal Outcome	Repair
Present case	2021	15	CD scar	1	CD	34	Increta	CD scar	No comp	Double-layer
Bouzari [34]	2019	26	Fundus	1	Myomectomy	-	No comp	-	No comp	-
Crespigny [35]	2019	23	Fundus ^†^	2	Spontaneous	30	CD	-	-	*
						23	Rupture	-	CP	Three-layer
						28	Percreta	Fundus	No comp	
Deka [10]	2011	16	Fundus	1	Spontaneous	32	Percreta	Fundus	No comp	Three-layer
Usta [8]	2007	17	Transverse	1	-	38	No comp	-	No comp	-

^†^ The site of uterine rupture in the 1st pregnancy is unknown. * The repair for first uterine rupture was not clarified. Abbreviations: Timing, timing of uterine rupture in previous pregnancy; site, site of uterine rupture in previous pregnancy; No.; number of cases; CD, cesarean delivery; GA, gestational age at delivery; PAS: placenta accreta spectrum; CP, cerebral palsy; no comp, no complication.

## Data Availability

All the studies included in this study are published in the literature.

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
