# Peer review of "Maternal and Fetal Outcomes after Prior Mid-Trimester Uterine Rupture: A Systematic Review with Our Experience"

_medicina, 2021, doi:10.3390/medicina57121294_

Round 1

Reviewer 1 Report

Dear Authors, well done! very complete update. If possible, add some information about repair, although there is not referred in the first trimester, like Palacios-Jaraquemada JM, Fiorillo A, von Petery F, Colaci D, Leguizamón G. Uterine repair and successful pregnancy after myometrial and placental rupture with massive haemoperitoneum. BJOG. 2009 Feb;116(3):456-60. doi: 10.1111/j.1471-0528.2008.01980.x. PMID: 19187380 or Belfort MA, Shamshirsaz AA, Cassady CI, Donepudi R, Espinoza J, Sanz Cortes M, King A, Nassr AA. Repair of a large uterine dehiscence during the second trimester leading to successful prolongation of the pregnancy. Am J Obstet Gynecol. 2020 Dec;223(6):929-932. doi: 10.1016/j.ajog.2020.07.037. Epub 2020 Jul 23. PMID: 32712004.

Many people don´t know about possibilities that could allow better neonatal resources. Beautiful work!

Best regards,

The reviewer

Author Response

Response to the Reviewers’ comments

We would like to thank the Editor and the Reviewers for the helpful comments. The following are our point-by-point responses to the comments along with descriptions if the revisions made to the manuscript. The revisions made to the manuscript are indicated using the “track changes” function of Microsoft Word.

Reviewer #1

Reviewer #1, comment #1

Dear Authors, well done! very complete update. If possible, add some information about repair, although there is not referred in the first trimester, like Palacios-Jaraquemada JM, Fiorillo A, von Petery F, Colaci D, Leguizamón G. Uterine repair and successful pregnancy after myometrial and placental rupture with massive haemoperitoneum. BJOG. 2009 Feb;116(3):456-60. doi: 10.1111/j.1471-0528.2008.01980.x. PMID: 19187380 or Belfort MA, Shamshirsaz AA, Cassady CI, Donepudi R, Espinoza J, Sanz Cortes M, King A, Nassr AA. Repair of a large uterine dehiscence during the second trimester leading to successful prolongation of the pregnancy. Am J Obstet Gynecol. 2020 Dec;223(6):929-932. doi: 10.1016/j.ajog.2020.07.037. Epub 2020 Jul 23. PMID: 32712004.

Many people don´t know about possibilities that could allow better neonatal resources. Beautiful work!

Best regards,

The reviewer

Reply: Table 1, lines 197-199, lines 299-319

Thank you for your insightful comments. According to the reviewer’s suggestion, we have added the information about the repair of mid-trimester uterine rupture to Table 1 and the main text (lines 198-200). We have also discussed uterine repair for mid-trimester uterine rupture and cited the suggested study (lines 299-319).

Reviewer 2 Report

This is a clinically significant review in which the outcomes in subsequent pregnancy following uterine rupture in mid-trimester is discussed in detail.

There are one thing to check.

L254 “We hypothesized that uterine rupture at the fundus possibly leads to a high rate of PAS.”

Is the site of the first uterine rupture the same or different from the site of second uterine rupture and placenta accreta in subsequent pregnancy?

This information will be useful for the reader to know the areas that should be closely evaluated for imaging in subsequent pregnancy.

Author Response

Response to the Reviewers’ comments

We would like to thank the Editor and the Reviewers for the helpful comments. The following are our point-by-point responses to the comments along with descriptions if the revisions made to the manuscript. The revisions made to the manuscript are indicated using the “track changes” function of Microsoft Word.

Reviewer #2

This is a clinically significant review in which the outcomes in subsequent pregnancy following uterine rupture in mid-trimester is discussed in detail.

There are one thing to check.

L254 “We hypothesized that uterine rupture at the fundus possibly leads to a high rate of PAS.”

Reply:

Thank you for your comments. We have revised the main text according to the reviewer’s suggestion. We hope that the reviewer will be satisfied with the revised manuscript.

Reviewer #2, comment #1

Is the site of the first uterine rupture the same or different from the site of second uterine rupture and placenta accreta in subsequent pregnancy?

Reply: Table 1

Thank you for your helpful comments. Since our description of the first uterine rupture site was not sufficiently clear, we have added this information to Table 1.

Reviewer #2, comment #2

This information will be useful for the reader to know the areas that should be closely evaluated for imaging in subsequent pregnancy.

Reply: Table 1, lines 203-204

We appreciate your valuable comment. Per the reviewer’s suggestion, we have added the site of placenta accreta spectrum to Table 1. The site of uterine rupture was also clarified, which was the same as the site of PAS in all cases (lines 203-204). As the reviewer pointed out, this information would be useful for the readers.